# In-vitro accuracy of casts for orthodontic purposes obtained by a conventional and by a printer workflow

**Sven Reich[1]\*, Hannah Herstell[1], Stefan Raith[2], Christina Kühne[1], Saskia Berndt[1]**

1 Department of Prosthodontics and Biomaterials, RWTH Aachen University Hospital, Aachen, Germany,
2 Department of Oral and Maxillofacial Surgery, RWTH Aachen University Hospital, Aachen, Germany

\* sreich@ukaachen.de

**Data Availability Statement:** All figure files are available from the https://figshare.com/s/0bbb1c2993794d67fdc2 database.

## Abstract

This *in-vitro* study was designed to investigate whether conventionally produced casts and printed casts for orthodontic purposes show comparable full-arch accuracy. To produce casts, either a conventional impression or a digital data set is needed. A fully dentate all ceramic master cast was digitized with an industrial scanner to obtain a digital reference cast [REF]. Intraoral scans [IOS] and alginate impressions were taken from the master cast so that ten printed and ten gypsum casts were obtained. The printed casts [DLP] were digitized by an industrial scanner and as well as the gypsum casts [GYPSUM]. The following absolute mean trueness evaluations by superimposition were accomplished: [REF vs. GYPSUM]; [REF vs. DLP]; [REF vs. IOS]; [IOS vs. DLP]. For precision analysis the data sets of [GYPSUM], [IOS] and [DLP] were available. The absolute mean trueness values were 68 µm ± 15 µm for [REF vs. GYPSUM], 46 µm ± 4 µm for [REF vs. DLP], 20 µm ± 2 µm for [REF vs. IOS] and 41 µm ± 4 µm for [IOS vs. DLP]. [REF vs. GYPSUM] and [REF vs. DLP], [REF vs. IOS], [REF vs. DLP] and [IOS vs. DLP] showed statistically significant differences. The precision values were 56 µm ± 17 µm for [GYPSUM], 25 µm ± 9 µm for [DLP] and 12 µm ± 2 µm for [IOS] and differed significantly among each other. In the present study the print workflow revealed superior results in comparison to the conventional workflow. Due to contrary deviations in the [REF vs. IOS] and the [IOS vs. DLP] data sets the overall trueness deviations was enhanced.

## Introduction

In orthodontics physical casts are still required for diagnosis and for the fabrication of appliances and retainer therapy [1, 2]. To produce casts, either a conventional impression or a digital data set is needed. Conventional impression taking often causes patient discomfort [3, 4]. Glisic et al. found that intraoral optical impressions were perceived as more pleasant regarding comfort, gag reflex, breathing, smell/sound, taste, and anxiety compared to alginate impressions [5].

The advantages of the digital workflow based on intraoral digital scans are unlimited replica production, less fracture risk and light weight of the produced casts [6]. To transfer a digital

**Funding:** Sven Reich received external funding by Sirona Dental Systems GmbH. Due to the fact that the funder is a company no official grant number is existent. For admistrative reasons only an internal account number is available. The coauthors were partially financed by this project. URL: https://www.dentsplysirona.com/de-de The funders didn't play any role in the study design, data collection and analysis, decision to publish or preparation of the manuscript.

**Competing interests:** The author S.R. (Sven Reich) has or had commercial relationships with the following companies including consulting, presentation activities, and third party-funded research: 3M OralCare, 3Shape, Amann Girrbach, Camlog, Oral Reconstruction Foundation, DCS, Dentaurum, Dentsply Sirona, Ivoclar Vivadent, Straumann, Vita Zahnfabrik. Sa.B., Ha.H., and S.R. (Stefan Raith) and CH.K. declare no conflict of interest.

data set of an intraoral optical impression into a physical cast, subtractive or additive methods can be used. Subtractive methods, such as milling, are less applicable due to time consumption and material waste [7]. Additive manufacturing, on the other hand, is characterized by less material waste and the possibility to produce more complex structures compared to subtractive methods [3]. It is represented by a variety of technologies such as vat polymerization, material extrusion as well as material jetting (MJ), binder jetting (BJ), and powder bed fusion [8, 9]. Vat polymerization comes with stereolithography (SLA), digital light processing (DLP), and continuous liquid interface production (CLIP) and describes the curing of a photosensitive polymer layer by layer using light [8–11]. DLP, as the investigated printer uses, polymerizes one layer at a time by exposure to light [11, 12]. Usually light-emitting diodes with wavelength ranges from deep ultraviolet to visible are used [11]. SLA cures one layer point-by-point with laser beams, mostly in the ultraviolet spectrum [10, 11, 13]. SLA printers are known to be used by private practice orthodontists to produce models in-house, for example to use them for aligner fabrication or retainer fabrication [1]. Other frequently used technologies in orthodontics are material extrusion with fused filament fabrication (FFF) and material jetting (MJ) [13]. The term material jetting (MJ) refers to a process in which an object is build up layer by layer by selectively jetting liquid polymers out of hundreds of nozzles and curing them with UV light [14]. Fused filament modelling (FFF), also known as fused deposition modelling (FDM), works by the extrusion of a thermoplastic material through a nozzle [10]. The nozzle heats up and deposits the material layer by layer [10].

The various indications for printed casts in orthodontics demand a certain full-arch accuracy. To determine accuracy, linear measurements or 3D surface comparisons can be approached. In this study, 3D surface comparisons were used. To allow surface comparisons, the printed test objects must be digitized [8]. The generated test data sets can then be registered either with the reference data set (trueness) or the test data sets are registered among themselves (precision). The registration, also referred to as superimposition, is performed by a best-fit algorithm orienting data sets in one coordinate system and deviations can be measured. When measuring, one data set is selected as reference and relative deviations between the two sets can be calculated. Comparing the surfaces, the test data set is positioned relatively above or under the reference's surface with resulting positive and negative deviation values. To achieve representative outcomes, absolute values must be used.

For cast accuracy, a clinical acceptable error range of $< 100$ μm to 500 μm was reported [8].

In this study, the full-arch accuracy of fully dentate gypsum and printed casts was evaluated with respect to the corresponding workflow. The null hypothesis stated was that gypsum and printed casts did not differ significantly in comparison to a reference cast at $p \leq 0.05$ with respect to accuracy. Furthermore, the differences within the digital workflow were investigated, comparing the digital reference cast file with the intraoral scans and the intraoral scans with the scans of the printed casts.

## Material and methods

To fabricate a ceramic maxillary reference cast a fully dentate typodont (Basic Study Model, KaVo Dental GmbH, Biberach, Germany) was digitized using an intraoral scanner (Cerec Primescan, software v5.1, Dentsply Sirona, Bensheim, Germany). The scan data were exported as a stereolithography (STL) file. A 3D software (Blender v2.78, Blender Foundation, Amsterdam, Netherlands) was used to divide the digital cast into an alveolar base and single teeth. With a 5-axis milling machine (Coritec 250i; imes-icore GmbH, Eiterfeld, Germany) the base was fabricated from yttrium-stabilized zirconium oxide (IPS e.max ZirCAD Prime, A3, Ivoclar Vivadent AG, Schaan, Liechtenstein). The teeth were produced from lithium disilicate glass-

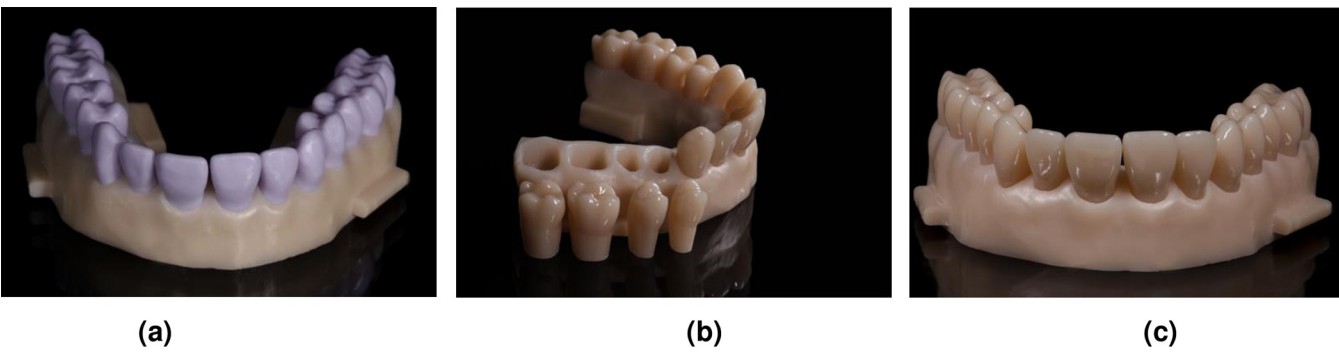

**Fig 1. Reference cast with an yttrium-stabilized zirconium oxide base and lithium disilicate glass-ceramic teeth.** (a) Lithium metasilicate (b) Glazed coronal surface (c) Reference cast.

ceramic blocks (IPS e.max CAD LT A2, Ivoclar Vivadent AG, Schaan, Liechtenstein) (Fig 1A) by another milling machine (Cerec MC XL, Dentsply Sirona, Bensheim, Germany). The lithium disilicate teeth were crystallized (Programat P500, Ivoclar Vivadent AG, Schaan, Liechtenstein), and coronally glazed (IPS Ivocolor Glaze Paste FLUO and mixing liquid alround, Ivoclar Vivadent AG, Schaan, Liechtenstein) (Fig 1B). For adhesive bonding between the base and teeth, the root segments were etched with fluoric acid and were silanized (IPS Ceramic Etching Gel and Monobond plus, Ivoclar Vivadent AG, Schaan, Liechtenstein). The sockets of the base were airborne particle abraded with $\leq$50 μm aluminum oxide at 0.1 MPa. Phosphoric acid containing composite (Panavia 21 Ex, Kuraray Europe GmbH, Hattersheim am Main, Germany) was applied as bonder.

To obtain a digital reference data set the cast (Fig 1C) was coated with scanning spray (Scan Spray Stone, Dentaco GmbH & Co. KG, Essen, Germany) and digitized with a high accuracy industrial structured light scanner (ILS) (GOM ATOS III Triple Scan, topometric GmbH, Göppingen, Germany). The reference cast was scanned five times to check for precision of the ILS. The scans were provided as STL files.

To simulate the conventional workflow, the interdental undercuts of the reference cast were blocked-out with wax. To check for wax residues the teeth's surfaces were examined with a microscope (ZEISS OPMI pico, Carl Zeiss Surgical GmbH). Alginate adhesive (Fix Tray Adhesive, Dentsply DeTrey GmbH, Konstanz, Germany) was applied to Rim-Lock trays, size U2 (Orbi-Lock, Orbis Dental). Alginate (23g/50ml water; mixed by Migma 200, MIKRONA GROUP AG, Schlieren, Switzerland) (ALGINoplast Fast Set, Kulzer GmbH) was used for impression taking. To avoid interaction between the alginate and glazed ceramic teeth, the cast was wetted with artificial saliva (Glandosane Spray, STADA Consumer Health & STADA-PHARM GmbH, Bad Vilbel, Germany). To ensure a standardized and evenly coating of the cast with the artificial saliva the cast was covered with two defined spray shots by the saliva applicator. As recommended in the manufacturer´s recommendations the impressions were instantly poured with Type IV gypsum (20ml water/100g gypsum; solidification time $\leq$ 30 min, setting expansion $\leq$ 0.15%, compressive strength $\geq$ 35 MPa) (Rapidur, DENTAURUM GmbH Co. KG) after 10 min disinfection (Kanipur, KANIEDENTA, Dentalmedizinische Erzeugnisse GmbH, Herford, Germany). The gypsum was vacuum mixed for 30 seconds and vibrated into the impressions. The casts were allowed to set for 30 minutes before separation from the impression. Sharp edges of the base were trimmed in dry condition, the teeth surfaces were not processed. This procedure was repeated ten times and the ten gypsum casts were digitized with an ILS.

For the digital workflow, digital optical impressions of the reference cast were made ten times by one investigator with an intraoral scanner (Primescan, Sirona Dental Systems GmbH, Bensheim, Germany) using the Cerec 5.2.2 RC 1 software. Scanning was performed following the scan path recommended by Passos et al. in a darkened room. In the acquisition phase, the casts were cut so that only relevant areas as teeth and adjacent gingiva were left. The digital casts were represented in model phase with their calculated bases, which were carved out to reduce printing material (Fig 2).

These datasets were imported as Sirona system specific ´cam´-files in the inLab CAM software (software 22.1 Beta 1, Sirona Dental Systems GmbH, Bensheim, Germany) and the printing process was executed by the DLP printer Primeprint (Sirona Dental Systems GmbH, Bensheim, Germany). The default settings for the support structures were accepted and "very high" detail level was selected. Additionally, the adjustment "optimized quality" influenced the ideal cast orientation on the building platform for precise additive manufacturing. During printing, two casts were fabricated simultaneously (Fig 3) using a specific resin (Primeprint Model, Sirona Dental Systems GmbH, Bensheim, Germany) as polymer material. After printing, the casts were transferred to the post processing unit (PPU, Sirona Dental Systems GmbH, Bensheim, Germany) for further polymerization and to remove any resin residues. After post-processing, the finished casts (Fig 4A and 4B) were detached from the building platform and the supports were removed. The printed casts were scanned by ILS to obtain STL-files.

For accuracy analysis, 3D surface comparison was performed by the use of the software GOM Inspect Suite 2020.

For trueness evaluation, one of the five STL files of the reference cast [REF] was randomly selected (Fig 5A). The ten gypsum and ten printed casts were represented by ten high-

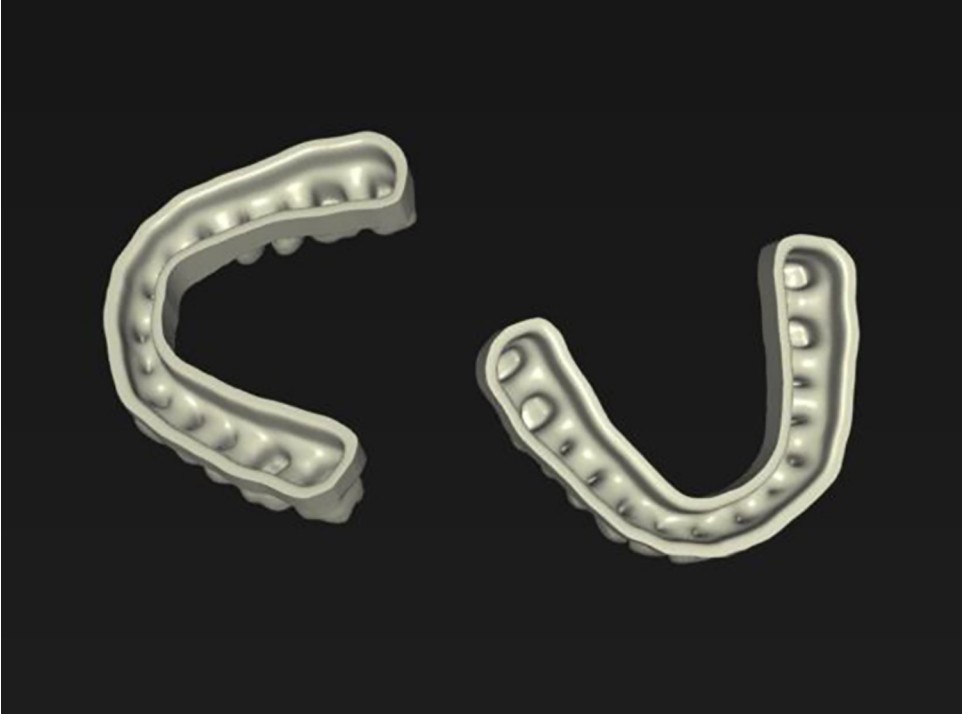

**Fig 2. Hollowed bases of two 'cam'-files.**

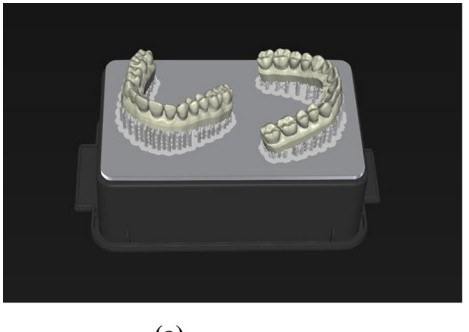
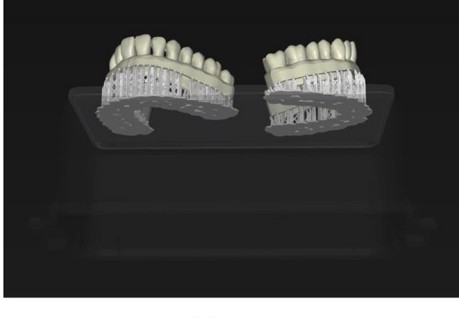

**(a)** **(b)**

**Fig 3.** (a) and (b) Arrangement of two simultaneously printed casts on the building platform.

resolution scans each ([GYPSUM], [DLP]) (Fig 5B). The ten intraoral optical impressions accomplished within the digital workflow were exported as STL files [IOS]. The combinations [REF vs. GYPSUM], [REF vs. DLP], [REF vs. IOS] and [IOS vs. DLP] were analyzed by super-imposition for trueness analysis using the software GOM Inspect Suite 2020. When the [REF] data set was involved [REF] was set as reference object. When [IOS] was compared with [DLP], [IOS] was chosen as reference object. Within the comparison [IOS vs. DLP], each [DLP] was compared to the corresponding [IOS].

Test object and reference objects were aligned in one common coordinate system via global best-fit alignment (Fig 5C). The used method for this automated alignment is a so called itera-tive closest point algorithm, that projects data points on the triangulated surface of test object toward their closed representation on the reference object [15]. These projections are used to transpose the test object closer to the reference object.to calculate a transformation matrix (consisting of translation and rotation) to match the test object closer to the reference surface. This procedure is done iteratively until no further improvement of the accordance between the two objects could be found. Subsequently, the registered meshes were cut identically in the same plane (Fig 5D).

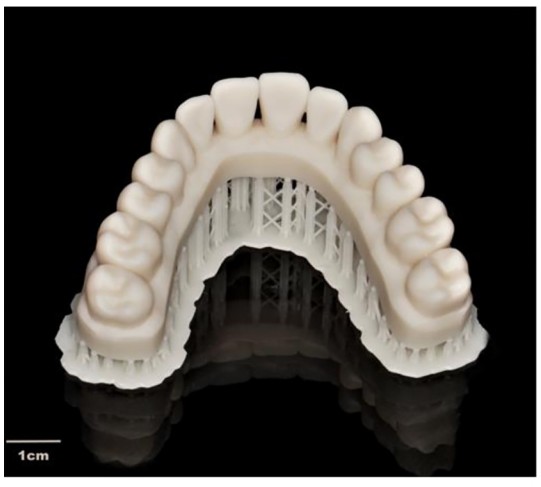
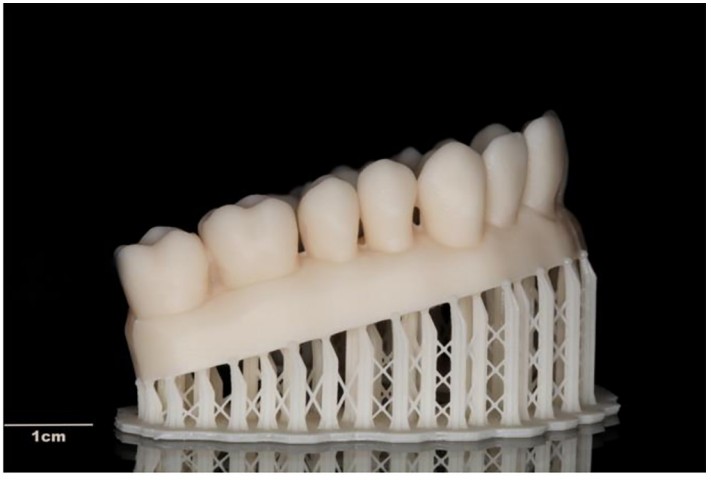

**(a)** **(b)**

**Fig 4.** (a) and (b) Printed cast with its attached support structures.

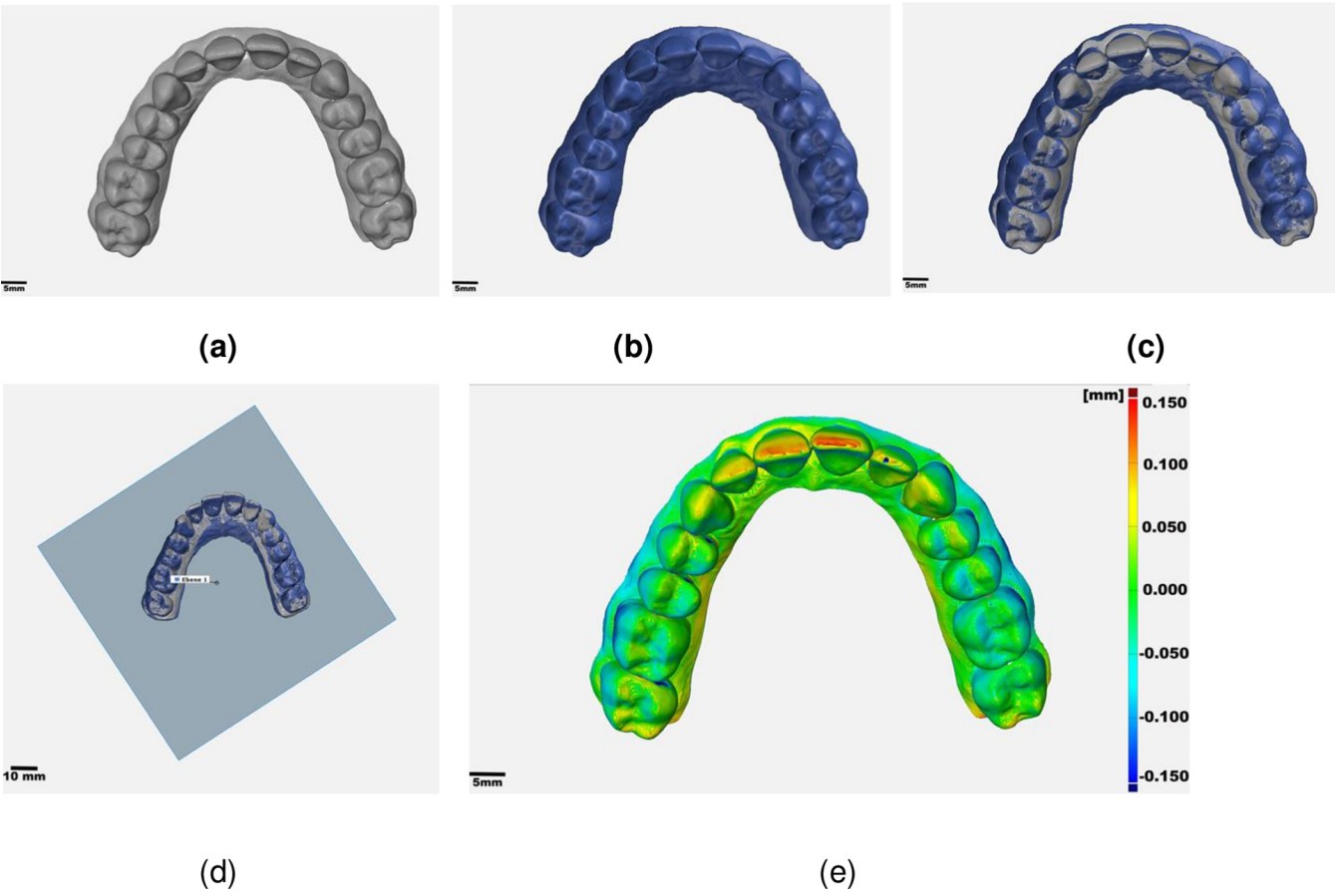

**Fig 5. Digital processing of the data sets in GOM Inspect 2020.** (a) reference cast; (b) test cast; (c) Best-fit alignment: superimposition of reference and test cast; (d) principle of trimming of all test casts; (e) surface comparison [Scale ± 150 μm].

For point-to-surface measurement (Fig 5E), the maximum tolerance was set to two millimeters and the surface comparison was applied on the test object.

All absolute values of each surface comparison between two scans, were executed in American Standard Code for Information Interchange (ASCII) file. One single data set included approximately 300,000 point-to-surface measurements. For automated data evaluation, a self-programmed python script was used, allowing the inclusion of all measured values or the exclusion of the highest and lowest 1%, 5%, or 10%. In the present study, the 99/1 quantile was used for calculating the absolute mean deviation, the root mean square error (RMSE), median, standard deviation, and the minimal and maximal distances. The RMSE describes the dispersion around a true value with respect to trueness and precision. RMSE is defined as:

$$RMSE = \sqrt{\frac{1}{n}\sum_{i=1}^{n}(x_i - \mu_i)^2}$$

$x_i$: predicted values of the reference cast, $\mu_i$: observed values of the test cast; n: number of observations

Precision was calculated applying the same registration and measuring strategy but comparing the high-resolution scans of the ten gypsum casts [GYPSUM], the ten printed casts [DLP] and the ten intraoral scans [IOS] within each group with each other. In consequence,

45 superimpositions and surface comparisons were performed in each group. The high-resolution scans of the reference cast [REF] were analyzed for precision, too (n = 5).

Statistical analysis was performed by using SPSS 26 (IBM Corporation, Armonk, NY, USA). The Shapiro-Wilk test revealed that the mean trueness data of [REF vs. GYPSUM] were not normally distributed. Therefore, the Mann-Whitney-U test (two-tailed) was used to test for a statistically significant difference between [REF vs. GYPSUM] and [REF vs. DLP] at p≤0.05. To analyze the digital workflow the t-test was applied (Bonferroni corrected) to check whether there was a significant difference between the data of the related samples [REF vs. IOS], [IOS vs. DLP]and [REF vs. DLP] at p≤0.05. The RMSE trueness data of [REF vs. GYPSUM], [REF vs. DLP], [REF vs. IOS] and [IOS vs. DLP] were normally distributed. The t-test was applied to test statistically significant differences.

For the mean precision data, the Shapiro-Wilk test revealed a non-normal distribution of [GYPSUM] and [DLP]. The Mann-Whitney-U test was used to test for a statistically significant difference between [GYPSUM] and [DLP]. To test for significant differences in [DLP] and [IOS] the Wilcoxon test was used.

For the precision based on RMSE, [GYPSUM] and [DLP] were not normally distributed. The Mann-Whitney-U test for independent and the Wilcoxon-W test for connected samples were performed ($\alpha$ = .05).

## Results

### 1. Trueness

The trueness data of [REF vs. GYPSUM], [REF vs. DLP], [REF vs. IOS] and [IOS vs. DLP] are displayed in Table 1.

**1.1 Absolute mean deviation.** The absolute mean deviation values were 68 μm (SD ± 15 μm) for [REF vs. GYPSUM], 46 μm (SD ± 4 μm) for [REF vs. DLP], 20 μm (SD ± 2 μm) for [REF vs. IOS] and 41 μm (SD ± 4 μm) for [IOS vs. DLP]. The values of [REF vs. GYPSUM] differed significantly from the values of [REF vs. DLP]. Among the digital workflow the result of [REF vs. DLP], [REF vs. IOS] and [IOS vs. DLP] differed significantly.

**1.2 Root mean square error.** The RMSE values were 122 μm (SD ± 12 μm) for [REF vs. GYPSUM], 78 μm (SD ± 3 μm) for [REF vs. DLP], 29 μm (SD ± 4 μm) for [REF vs. IOS] and 54 μm (SD ± 5 μm) for [IOS vs. DLP]. The RMSE values of [REF vs. GYPSUM] differed significantly from the values of [REF vs. DLP]. [REF vs. DLP], [REF vs. IOS] and [IOS vs. DLP] differed significantly.

### 2. Precision

The precision data of [GYPSUM], [DLP], and [IOS] are shown in Table 2.

**2.1. Absolute mean deviation.** The precision based on absolute mean deviation values was 56 μm (SD ± 17 μm) for [GYPSUM]; 25 μm (SD ± 8 μm) for [DLP] and 12 μm

**Table 1. Trueness values.**

|  | N | Absolute Mean Deviation [μm] | | | | RMS Error [μm] | | | |
|---|---|---|---|---|---|---|---|---|---|
|  |  | Minimum | Maximum | Mean ± Standard Deviation | Median | Minimum | Maximum | Mean ± Standard Deviation | Median |
| [REF vs. GYPSUM] | 10 | 54 | 106 | 68± 15 | 63 | 105 | 147 | 122± 12 | 122 |
| [REF vs. DLP] | 10 | 39 | 52 | 46± 4 | 46 | 81 | 69 | 78±3 | 88 |
| [REF vs. IOS] | 10 | 18 | 22 | 20± 2 | 21 | 25 | 38 | 29± 4 | 29 |
| [IOS vs. DLP] | 10 | 33 | 46 | 41± 4 | 42 | 44 | 60 | 54±5 | 54 |

**Table 2. Precision values.**

| | N | Precision based on Absolute Mean Deviation [μm] | | | | Precision based on RMS Error [μm] | | | |
|---|---|---|---|---|---|---|---|---|---|
| | | Minimum | Maximum | Mean ± Standard Deviation | Median | Minimum | Maximum | Mean ± Standard Deviation | Median |
| [GYPSUM] | 10 | 36 | 94 | 56± 17 | 50 | 60 | 132 | 86± 21 | 75 |
| [DLP] | 10 | 0 | 45 | 25± 8 | 23 | 0 | 36 | 22± 6 | 20 |
| [IOS] | 10 | 9 | 15 | 12± 2 | 12 | 15 | 24 | 19± 2 | 19 |

(SD ± 2 μm) for [IOS]. The comparisons of [GYPSUM] and [DLP] and of [IOS] and [DLP] showed statistically significant differences.

**2.2. Root mean square error.** The precision based on RMSE values was 86 μm (SD ± 21 μm) for [GYPSUM], 22 μm (SD ± 6 μm) for [DLP] and 19 μm (SD ± 2 μm) for [IOS]. The comparisons of [GYPSUM] and [DLP] and of [IOS] and [DLP] showed statistically significant differences.

**2.3. Atos scans.** The precision of the five ATOS scans was 3 μm (SD ± 2 μm).

## Discussion

The null hypothesis that there was no statistically significant difference in accuracy between gypsum and printed casts was rejected.

Given absolute mean trueness deviations of 68 μm (SD± 15 μm) for [REF vs. GYPSUM] and 46 μm (SD± 4 μm) for [REF vs. DLP], and an absolute mean precision of 56 μm (SD± 17 μm) for [GYPSUM] and 25 μm (SD± 9 μm) for [DLP], the potential sources of inaccuracies must be discussed. Within the conventional workflow, the alginate was used as standard material for impression taking in orthodontics according to the manufacturer's instructions. Alginate impressions involve the risk of dimensional deformation because of moisture absorption or moisture loss, and the risk of permanent deformation due to undercuts (e.g., orthodontic brackets) [16]. Combined with the gypsum processing, the multistep workflow can accumulate errors [17]. The optical impressions provide the basis of the digital workflow. They can be influenced by the underlying scanning technology, the ambient conditions like optical properties of the surface to be scanned and natural and external light sources, scanning strategy, and data processing (alignment algorithms) [17, 18]. In the present study, the trueness for [REF vs. IOS] was 20 μm (SD± 2 μm) and the precision was 12 μm (SD± 1 μm) for [IOS]. These results were in accordance with previous studies. The same working group found trueness values of 19 μm (SD ± 3 μm) and precision values of 15 μm (SD ± 4 μm) in another study using the same intraoral scanner and software version [19]. A full-arch trueness of 29 μm (SD ± 3 μm) and precision of 15 μm (SD ± 5 μm) was achieved with an older software version (v5.1) [20]. Schmidt et al. obtained mean trueness values of 33.8 μm (SD ± 31.5 μm) and Ender et al. reported a trueness value of 33.9 μm (SD ± 7.8 μm) and a precision value of 31.3 μm (SD ± 10.3 μm) for the (90–10)/2 percentile [21, 22].

During computer-aided manufacturing, software related inaccuracies may occur, next to inaccuracies caused by the polymerization shrinkage and forces excerted on the object when removing printed layers from the vat, layer thickness, and post processing. Post processing includes the rinse off of resin residues and further polymerization. Incorrect post processing can result in accumulated resin, leading to material excess. The percentage of shrinkage during polymerization is a key factor for 3D-deviations. An absolute mean trueness of [IOS vs. DLP] with 41 μm (SD± 4 μm) reflects the beforementioned factors. Whereas for [REF vs. IOS] a widening of the arch in the molar area was revealed, for [IOS vs. DLP], a relative compression of the arch was found. Thus, the overall workflow compensated trueness deviations (Fig 6).

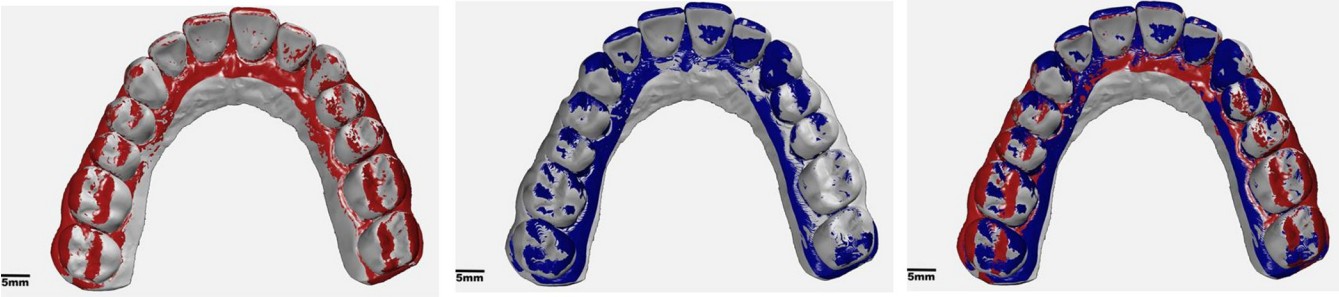

**Fig 6.** Compensation of the arch expansion during the entire workflow: (a) [REF vs. IOS] (grey = reference cast; red = IOS) The arch of [IOS] is widened in relation to the reference cast; (b) [REF vs. DLP] (grey = ref; blue = printed) [DLP] shows a narrower arch than [REF]; (c) Superimposition of [REF vs. IOS] and [REF vs. DLP] The expansion of the intraoral scan is overcompensated during the printing process.

The phenomenon of the widening of the arch for [REF vs. IOS] in the molar region could be explained by stitching errors. Due to the fact, that a 3D full arch impression has to be built up by a number of single 3D-images, these single exposures have to be joined together by superimposition/registration of identical structural parts. In general, the more individually structured the scanned surface the more accurate is the registration. A proper registration is the prerequisite for a good full-arch accuracy. On the other hand, the scan procedure of the anterior teeth, especially the lower ones, is prone to errors by the stitching algorithms, because they are characterized by flat and steep facets and narrow incisal plateaus [20].

Other studies evaluating the accuracy of full arch DLP printed dental casts described absolute mean deviation trueness values of 97μm (SD± 73μm) for hollowed bases, 87 μm (SD± 57μm) for solid bases, and 106 μm (SD± 23 μm) as RSME emphasizing outliers [23, 24]. Zhang et al. investigated trueness with regard to different layer thicknesses [25]. Three DLP (Evo-Dent, EncaDent, Vida HD) printers and one SLA printer (Form 2) were included (25). 50 μm and 100 μm layer thicknesses were applied for all printer types, one DLP printer additionally allowed layer thicknesses of 20 or 30 μm, while the SLA printer added 25 μm [25]. The highest accuracy with absolute mean deviation values of 23 μm (SD± 31 μm) in the maxilla and 30 μm (SD± 43 μm) in the mandible were found for the EvoDent printer at 50 μm layer thickness [25]. The lowest accuracy was shown by the Form 2 printer at 100 μm layer thickness with mean absolute deviation of 51 μm (SD± 71 μm) in the maxilla and 57 μm (SD± 83 μm) in the mandible [25]. For DLP technology, 50 μm was the optimum layer thickness [25]. For SLA technology, decreasing layer thickness resulted in higher accuracy [25]. Primeprint allows layer thicknesses from 50 up to 200 μm to allow faster printing with increased layer thicknesses. For our investigations, a layer thickness of 50 μm was used to ensure the highest possible detail level. Primeprint as a printer especially for dental indications applying the DLP technique may differ in print quality from more affordable options due to high-end equipment and settings.

For the precision of casts printed with DLP technology, studies described a mean RMSE of 53 μm (SD± 18 μm) and 76 (SD± 14 μm) [24, 26]. The precision in the present study with an absolute mean value of 25 μm and a mean RMSE of 22 μm were below these values.

The present study revealed that it was possible to fabricate printed casts with adequate full arch accuracy for orthodontic purposes. The results achieved by the printer-workflow were better than the simulated conventional workflow under in-vitro conditions.

In orthodontics a deviation of < 300 μm for gypsum or printed casts from the reference was determined to be clinically acceptable [8, 23]. Compared to casts used in prosthodontics these requirements are less strict [8, 23]. Thresholds of <100 μm mean trueness values are

considered as highly accurate and acceptable for the clinical workflow in fixed and implant prosthodontics [8, 23]. Based on these findings, the accuracy of the investigated gypsum and printed casts can be considered as clinically acceptable for orthodontic purposes.

## Limitations

This in-vitro study design differs from the in-vivo procedures in some aspects. For the digital workflow, the cast was scanned in an ideal environment. In a clinical situation, intraoral conditions such as saliva, humidity, patient movement and lack of space may influence the accuracy of intraoral scans [27]. When the conventional impression taking was simulated, the reference cast was wetted with artificial saliva. The artificial saliva was used to reduce the adhesion of the alginate material on the dry surface of the reference cast. Due to the fact that the surface for the optical impression should be free of saliva during that workflow, no artificial saliva was applied. Prior to alginate impression taking, interdental undercuts were blocked out with wax to avoid tearing out the interdental spaces which might compromise the comparability to actual clinical procedures [28]. This was not necessary for the simulation of intraoral scanning procedure. Furthermore, alginate impressions are likely to be poured after a certain storage time with varying storage conditions in a clinical scenario. This might result in volume changes and distortion of the impression material and therefore in inaccuracies [28].

To minimize the risk of volumetric changes in this study, the impressions were poured directly after a disinfection time of 10 minutes.

## Conclusion

Within the limits of this study, the following conclusions were drawn:

1. Under in-vitro conditions the print workflow using a DLP printer revealed significantly better trueness and precision values compared to the alginate/gypsum-based workflow with respect to absolute mean and mean RMSE results.

2. The results were competitive to the present literature.

3. Both, the gypsum and the 3D printed cast, produced acceptable accuracy.

4. Due to contrary deviations in the [REF vs. IOS] and the [IOS vs. DLP] data sets, a compensation of overall trueness deviations is observed.

## Acknowledgments

The study was supported by our colleague Berfin Yatmaz.

## Author Contributions

**Conceptualization:** Sven Reich, Hannah Herstell, Stefan Raith, Saskia Berndt.

**Data curation:** Hannah Herstell, Saskia Berndt.

**Investigation:** Sven Reich, Hannah Herstell, Saskia Berndt.

**Methodology:** Sven Reich, Hannah Herstell, Stefan Raith, Saskia Berndt.

**Project administration:** Sven Reich.

**Supervision:** Sven Reich, Stefan Raith.

**Visualization:** Hannah Herstell, Christina Kühne, Saskia Berndt.

**Writing – original draft:** Sven Reich, Hannah Herstell, Saskia Berndt.

**Writing – review & editing:** Sven Reich, Hannah Herstell, Stefan Raith, Saskia Berndt.

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
