## [Decision Letter · Decision Letter 0]

6 Jul 2022

PONE-D-22-10316In-vitro accuracy of casts for orthodontic purposes obtained by a conventional and by a printer workflowPLOS ONE

Dear Dr. Berndt,

Thank you for submitting your manuscript to PLOS ONE. After careful consideration, we feel that it has merit but does not fully meet PLOS ONE’s publication criteria as it currently stands. Therefore, we invite you to submit a revised version of the manuscript that addresses the points raised during the review process.

We look forward to receiving your revised manuscript.

Kind regards,

Mirza Rustum Baig

Academic Editor

PLOS ONE

Journal Requirements:

2. Please ensure that you include a title page within your main document. You should list all authors and all affiliations as per our author instructions and clearly indicate the corresponding author.

Additional Editor Comments (if provided):

Thank you for the submission. The reviewers have raised some concerns. Please address these and resubmit the manuscript for further consideration.

Reviewers' comments:

Reviewer's Responses to Questions

**Comments to the Author**

1. Is the manuscript technically sound, and do the data support the conclusions?

Reviewer #1: Partly

Reviewer #2: Yes

Reviewer #3: Partly

Reviewer #4: Yes

2. Has the statistical analysis been performed appropriately and rigorously? 

Reviewer #1: Yes

Reviewer #2: Yes

Reviewer #3: I Don't Know

Reviewer #4: Yes

3. Have the authors made all data underlying the findings in their manuscript fully available?

Reviewer #1: Yes

Reviewer #2: No

Reviewer #3: No

Reviewer #4: Yes

4. Is the manuscript presented in an intelligible fashion and written in standard English?

Reviewer #1: Yes

Reviewer #2: Yes

Reviewer #3: No

Reviewer #4: Yes

5. Review Comments to the Author

Reviewer #1: There are fundamental issues with the design that need to be addressed in the writing. The conventional alternative method to introral scanning is to use PVS impression material. The use of alginate impressions was never the accepted standard. The manuscript has to be rewritten to emphasize this. Unfortunately with the potential conflict of interest, the fact that the design favors intraoral scanners is problematic. If the authors really wanted to "in good faith" test the conventional system they should have gone with PVS, not alginate. Accordingly, the paper should highlight this issue and downgrade any conclusions. Also, conventional impression are typically not poured to fabricate aligners. They are usually scanned in impression form OR scanned for inclusion in a digital workflow. Honestly, I was on the fence about recommending "major rewrite" OR "reject", mainly because of the choices in the design.

On the minor side of notes, the English is acceptable, but could use some editing. The introductions goes a lot into some 3D printing technologies that was not actually used in the study nor is in common clinical use.

Reviewer #2: Dear Authors,

The aim of this study was to evaluate whether conventionally produced casts and printed casts for aligner therapy show comparabel full-arch accuracy. While the topic is fitting to the journal scope, some concerns were raised to publish as a scientific paper. Revise the manuscript by following comments.

Major points

All figures were not visible. Please make sure the link to download figures.

Materials and methods

"best-fit alignment" is a function of CAD software and black box. Add a brief explanation for this alignment.

Ten times measurement for same specimen is not fit to statistical analysis. The specimens should be different on each measurement.

The Discussion section is short and should be enhanced.

Minor points

Abstract

"in-vitro" should be described in Italic font. Make sure all related points and revise them.

"xx um (SD+-xx um)" should be simplified to "xx +- xx um".

Materials and methods

"a standard tessellation language" should be modified to "a stereolithography".

A later standard tessellation language could be described as STL.

".,." is a typo.

Some company names appeared many times after the first description.

"stl" should be modified to "STL".

"300.000" might be modified to "300,000".

MAE might be used as a candidate evaluation criteria.

Reviewer #3: This was an in vitro study to compare conventional and digital/printed models.

Abstract: I find that there were too much abbreviations that it took me a long time to work out what the abstract was describing. I think the authors should limit the abbreviations and formulas to the content of the manuscript to enable readers to understand what the whole text would be about.

Manuscript

I think the author should consider the clinical importance of their study. Clinically acceptable threshold should be explained early in the text. The authors should consider if the study is really about the superiority of the printed models or clinically acceptability?

What would be the added value for the more complex digital workflow which may be more costly than conventional, when the latter is already practical. Furthermore, if the machines are for industrial, how can the clinicians relate as what's accessible to them are commercial machines?

Given that the study is only about DLP printer, it would benefit the reader that a general term use for [PRINT] be more specific to DLP. This is now relevant because other types of 3D printers are so easily accessible that readers would now want to focus which printer the study is about.

for ease of readership, I would suggest for the authors to minimise using formulas in text and describe their work.

The conclusion should reflect the outcome based on the aims of the study.

Reviewer #4: Accuracy of conventional and printed casts for orthodontic purposes

General

• This is a well written report.

• The study was thoroughly carried out.

TITLE

Adequate

ABSTRACT

Adequate

INTRODUCTION

Adequate

MATERIALS AND METHODS

Materials & Instruments:

1. Clarify if the reference cast was scanned and digitized 5 times (line no.89) or 10 times (line no.186).

2. Please describe further details of the standardization of application of artificial saliva (especially in relation to the amount) before taking alginate impression (line no. 104).

3. As described in Discussion (line no.) which states the moisture sensitive property of alginate, report on the time taken after impression until casting and whether moisture loss from alginate impression was minimized similar to normal clinical situation.

RESULTS

4. Suggest reporting on precision analysis results of the high-resolution scans of the reference cast [REF] (line no.185).

DISCUSSION

5. Discuss the widening of the arch in the molar area with intraoral scanning.

LIMITATIONS

6. Limitations need to be addressed, comment on inherent weakness of in vitro studying, focusing on possible differences in clinical situation. For example, intraoral scanning of the reference cast for the digital workflow was done on dry cast which may not truly reflect oral condition, and the interdental undercuts of the reference cast was blocked out with wax before alginate impression and this may not be standard clinical practice.

CONCLUSION

Suggest using the term ‘superior precision results’ to be more specific to the study findings.

ACKNOWLEDGEMENT

Suggest indicating if Berfin Yatmaz is a person or a company

6. PLOS authors have the option to publish the peer review history of their article (what does this mean?). If published, this will include your full peer review and any attached files.

Reviewer #1: No

Reviewer #2: No

Reviewer #3: No

Reviewer #4: **Yes: **MANG CHEK WEY

---

## [Author Response · Author response to Decision Letter 0]

27 Aug 2022

Dear reviewer

Thank you very much for the revision of our manuscript entitled ‘In-vitro accuracy of casts for orthodontic purposes obtained by a conventional and by a printer workflow - Accuracy of conventional and printed casts for orthodontic purposes’.

Attached you will find the item-by-item response.

Thank you very much for the over-all assessment.

---

## [Decision Letter · Decision Letter 1]

2 Oct 2022

PONE-D-22-10316R1In-vitro accuracy of casts for orthodontic purposes obtained by a conventional and by a printer workflowPLOS ONE

Dear Dr. Berndt,

Thank you for submitting your manuscript to PLOS ONE. After careful consideration, we feel that it has merit but does not fully meet PLOS ONE’s publication criteria as it currently stands. Therefore, we invite you to submit a revised version of the manuscript that addresses the points raised during the review process.

Kindly address the concerns raised by the reviewers. I look forward to receiving your revised version of the manuscript. 

We look forward to receiving your revised manuscript.

Kind regards,

Mirza Rustum Baig

Academic Editor

PLOS ONE

Reviewers' comments:

Reviewer's Responses to Questions

**Comments to the Author**

1. If the authors have adequately addressed your comments raised in a previous round of review and you feel that this manuscript is now acceptable for publication, you may indicate that here to bypass the “Comments to the Author” section, enter your conflict of interest statement in the “Confidential to Editor” section, and submit your "Accept" recommendation.

Reviewer #1: (No Response)

Reviewer #2: (No Response)

2. Is the manuscript technically sound, and do the data support the conclusions?

Reviewer #1: Partly

Reviewer #2: Yes

3. Has the statistical analysis been performed appropriately and rigorously? 

Reviewer #1: Yes

Reviewer #2: Yes

4. Have the authors made all data underlying the findings in their manuscript fully available?

Reviewer #1: Yes

Reviewer #2: Yes

5. Is the manuscript presented in an intelligible fashion and written in standard English?

Reviewer #1: Yes

Reviewer #2: Yes

6. Review Comments to the Author

Reviewer #1: Unfortunately, in their response to comments, the authors have decided to dodge some concerns or flat out double down on their shortcomings in the study. There seems to be some confusion in the their arguments against the "conventional workflow". Nowadays, even when physical impressions are made for the sake of aligner fabrication, gypsum casts are seldom used. Yes, in orthodontics we still typically use stone casts for diagnosis, retainer fabrication etc, but who produces a series of gypsum casts for aligner fabrication?? This is obviously the case because it would be pointless to digitally "move" the teeth and create stages then go back to gypsum casts. Once in the digital format, it is typically printed. Regardless, the study is slightly improved compared to last version. I still have some feedback and comments below. With minor tweaks, it will be more acceptable.

1)The IOS and alginate impressions were obtained under different circumstances. It would have been very easy for the authors to use artificial saliva and wax block-out for the IOS too. This should be highlighted in the limitations

2) The authors need to report the details of the type and properties of the "gypsum used" as different types of dental stones have different properties.

3) DLP printers differ significantly in print quality. The high-end printer and resin used might not be completely comparable to lower-end or more affordable options. This no way a criticism of the study, just a mere comment.

4) Was it not possible for the authors to directly compare DLP vs. Gypsum in table 1

5) In the discussion, line 285, it is not relevant to the study to mention brackets. As stated by the authors, the study is about aligners, which makes brackets irrelevant.

6)Tthe conclusions need some editing. Point 1 and 2 of the conclusions can be merged together.

7) Replace the term "conventional" with "alginate-based" OR "alginate/gypsum" in the conclusions.

Reviewer #2: Dear Authors,

The manuscript was mostly revised according to the reviewer's comments. However, all figures were still not visible from the embedded link. The link has been expired. Please put all figures in the main text.

7. PLOS authors have the option to publish the peer review history of their article (what does this mean?). If published, this will include your full peer review and any attached files.

Reviewer #1: No

Reviewer #2: No

---

## [Author Response · Author response to Decision Letter 1]

17 Nov 2022

Dear reviewer

Thank you very much for the revision of our manuscript entitled ‘In-vitro accuracy of casts for orthodontic purposes obtained by a conventional and by a printer workflow - Accuracy of conventional and printed casts for orthodontic purposes’.

Attached you will find the item-by-item response.

Thank you very much for the over-all assessment.

---

## [Decision Letter · Decision Letter 2]

24 Feb 2023

In-vitro accuracy of casts for orthodontic purposes obtained by a conventional and by a printer workflow

PONE-D-22-10316R2

Dear Dr. Berndt,

We’re pleased to inform you that your manuscript has been judged scientifically suitable for publication and will be formally accepted for publication once it meets all outstanding technical requirements.

Kind regards,

Sompop Bencharit, DDS, MS, PhD, FACP

Academic Editor

PLOS ONE

Additional Editor Comments (optional):

The reviewer and editor were pleased with the revised manuscript. We appreciate your time and contribution.

Reviewers' comments:

Reviewer's Responses to Questions

**Comments to the Author**

1. If the authors have adequately addressed your comments raised in a previous round of review and you feel that this manuscript is now acceptable for publication, you may indicate that here to bypass the “Comments to the Author” section, enter your conflict of interest statement in the “Confidential to Editor” section, and submit your "Accept" recommendation.

Reviewer #2: All comments have been addressed

2. Is the manuscript technically sound, and do the data support the conclusions?

Reviewer #2: Yes

3. Has the statistical analysis been performed appropriately and rigorously? 

Reviewer #2: Yes

4. Have the authors made all data underlying the findings in their manuscript fully available?

Reviewer #2: Yes

5. Is the manuscript presented in an intelligible fashion and written in standard English?

Reviewer #2: Yes

6. Review Comments to the Author

Reviewer #2: Dear Authors,

The manuscript was appropriately revised according to the reviewer comment. Thanks for your effort.

7. PLOS authors have the option to publish the peer review history of their article (what does this mean?). If published, this will include your full peer review and any attached files.

Reviewer #2: No

---

## [Editor Report · Acceptance letter]

6 Mar 2023

PONE-D-22-10316R2 

In-vitro accuracy of casts for orthodontic purposes obtained by a conventional and by a printer workflow 

Dear Dr. Berndt:

I'm pleased to inform you that your manuscript has been deemed suitable for publication in PLOS ONE. Congratulations! Your manuscript is now with our production department. 

Kind regards, 

on behalf of

Dr. Sompop Bencharit 

Academic Editor

PLOS ONE